# Soil Microbes Mediate Productivity Differences Between Natural and Plantation Forests

**DOI:** 10.3390/plants15010098

**Published:** 2025-12-28

**Authors:** Xing Zhang, Mengya Yang, Yangyang Liu, Jinkun Ye, Jiechen Tangyu, Jie Gao, Weiguo Liu, Yuchuan Fan

**Affiliations:** 1College of Ecology and Environment, Xinjiang University, Urumqi 830046, China; zxyybh@163.com (X.Z.); ryry9912@163.com (J.Y.); tyjc0409111@163.com (J.T.); 2College of Life Sciences, Northeast Agricultural University, Harbin 150030, China; mengyayang1212@163.com; 3College of Life Sciences, Xinjiang Normal University, Urumqi 830054, China; lyy1366@163.com; 4Jiyang College, Zhejiang A&F University, Zhuji 311800, China

**Keywords:** soil microbial communities, forest productivity, microbial diversity and function, biotic–abiotic interactions

## Abstract

While climate is known to regulate forest productivity, the mechanistic contribution of soil microbial communities—and whether it differs between natural and plantation forests—remains poorly quantified at broad scales. Here, we provide a synthesis-level, unified analysis that jointly evaluates climate, edaphic conditions, and soil microbes to compare mechanistic pathways underlying productivity divergence between forest types. We synthesized 237 observations across China and integrated productivity metrics—gross primary productivity (GPP) and net primary productivity (NPP)—with microbial diversity, dominant taxa, and soil drivers to compare natural and plantation forests within the current environmental coverage. Plantation productivity showed nonlinear responses to microbial diversity and appeared more sensitive than natural forests. Natural forests exhibited higher bacterial Shannon and Chao1 but lower fungal Chao1 and were characterized by taxa such as *Nitrobacter*, *Bradyrhizobium*, and *Cortinarius*. In contrast, plantations were characterized by taxa often associated with disturbance tolerance and opportunistic life-history strategies (e.g., *Sphingomonas*, *Fusarium*, *Gemmatimonas*), consistent with potential functional simplification. Structural equation models identified climate as the strongest correlate of productivity, while soil properties showed contrasting associations with microbial diversity across forest types. Random forest models further highlighted cation-exchange capacity and total nitrogen as key predictors of microbial diversity in plantations. Overall, our results indicate that soil microbial communities are differentially associated with forest productivity across forest types and environmental contexts and underscore the need for future climate-comparable designs and management-intensity information to more robustly isolate microbial contributions.

## 1. Introduction

Forest ecosystems play a fundamental role in global carbon cycling, biodiversity conservation, and climate regulation. In the face of increasing anthropogenic pressures and climate variability, understanding the factors that control forest productivity has become a pressing scientific priority [1]. Traditionally, climatic variables such as mean annual temperature (MAT), precipitation (MAP), and potential evapotranspiration (PET) have been considered the dominant controls of forest productivity. These factors are known to influence plant physiological activity, leaf area index, and water use efficiency [2]. However, mounting evidence suggests that these abiotic parameters alone cannot fully explain the variability in productivity observed across different forest types and biogeographic zones [3].

Recent advances in microbial ecology have brought attention to the pivotal role of soil microbial communities in shaping forest ecosystem function. Soil microbes drive biogeochemical cycling, regulate soil organic matter decomposition, and facilitate nutrient availability to plants [4]. They also form symbiotic associations with roots that influence water uptake, disease resistance, and photosynthetic efficiency [5]. For example, ectomycorrhizal fungi and nitrogen-fixing bacteria can contribute to phosphorus mobilization and sustained nitrogen inputs, especially in nutrient-limited environments [6]. Despite these critical contributions, the extent to which differences in soil microbial communities between natural and plantation forests account for variation in forest productivity remains understudied.

Natural forests are typically characterized by diverse plant assemblages, intact litter layers, and relatively undisturbed soil profiles, creating favorable conditions for microbial colonization and community development. In such systems, microbial diversity and functional richness are generally high [7]. By contrast, plantations are often established as simplified stands and may experience recurrent management disturbances, which can alter microbial diversity, community structure, and functional composition compared with natural forests [8,9].

Plantation forests—often established for commercial timber production or ecological restoration—are generally managed as monospecific stands with simplified aboveground structures and repeated anthropogenic disturbance [10,11]. To interpret microbial community differences under such “disturbance–resource pulse” contexts, we adopt the ecological framework of *r*/*K* strategies to describe microbial life-history traits: *r*-selected taxa are typically fast-growing, highly colonizing, and disturbance-tolerant, yet often contribute less to long-term nutrient stabilization; by contrast, *K*-strategist taxa are slower-growing and resource-efficient microbes adapted to more stable environments, and are commonly linked to sustained nutrient cycling, organic matter turnover, and ecosystem resilience [10]. Under this framework, plantations may favor opportunistic taxa and exhibit reduced microbial network complexity, potentially weakening microbial support for efficient nutrient cycling and stable productivity [12]. Therefore, understanding how such strategy shifts in plantation microbiomes interact with climate and soil environments is essential for explaining productivity divergence across forest types.

Beyond microbial composition per se, interactions among climate, soil nutrients, and microbial communities are increasingly recognized as key regulators of forest productivity [13]. Soil cation exchange capacity (CEC), total nitrogen (TN), and pH can influence microbial activity and functional redundancy by shaping nutrient availability and the physicochemical environment, thereby altering the resource context available to plants. Importantly, accumulating evidence suggests that microbe–productivity linkages may differ systematically between natural and plantation forests: natural forests often show stronger associations between microbial diversity and productivity, whereas plantations tend to exhibit weaker or more context-dependent relationships, likely due to simplified belowground networks and stronger abiotic and management constraints [14]. Together, these observations motivate an integrated evaluation of climate–soil–microbe pathways to clarify how belowground communities contribute to productivity differences across forest types.

Despite these insights, a holistic understanding of the microbial mechanisms associated with productivity differences across forest types remains incomplete. Previous studies have rarely integrated climate, edaphic, and microbial factors within a single analytical framework, limiting our ability to evaluate pathways and develop ecosystem-specific management implications. To fill these gaps, the present study addresses the following research questions: (1) To what extent are differences in soil microbial diversity and community features associated with variation in gross and net primary productivity (GPP/NPP) between natural and plantation forests within the current environmental coverage? (2) Through which pathways do climate and soil properties relate to microbial diversity and forest productivity, and do these pathways differ between natural and plantation forests? Guided by these questions, we test the following hypotheses: H1, microbial diversity is more strongly and positively associated with productivity (GPP/NPP) in natural forests than in plantations within the current environmental coverage. H2, climate primarily sets the baseline productivity, while soil physicochemical and nutrient conditions shape microbial diversity and community structure; microbial diversity partially mediates soil–productivity linkages, and these mediation pathways differ between natural and plantation forests.

## 2. Results

### 2.1. Spatial Heterogeneity in Forest Distribution and Climatic Conditions

Natural and plantation forests in China exhibit pronounced disparities in spatial distribution (Figure 1a). Natural forests predominantly occupy mountainous regions and protected areas, spanning diverse geographic zones from northeastern and southwestern to southern China. In contrast, plantation forests are concentrated in economically developed, densely populated regions or areas undergoing active ecological restoration, such as east China, south China, and parts of central China.

To elucidate the climatic drivers of these distributions, we mapped forest types against a Whittaker biome framework using temperature and precipitation gradients (Figure 1b). Natural forests occupy broader climatic ranges, from cool, arid zones to warm, humid regions. Plantation forests, however, cluster in areas with optimal temperature and precipitation, reflecting management priorities for economic yield and restoration efficacy.

### 2.2. Asymmetric Relationships Between NPP and Microbial Indices in Natural vs. Plantation Forests

Nonlinear relationships between NPP and microbial diversity indices (Shannon and Chao1) differed markedly between forest types (Figure 2a,b). In plantation forests, NPP initially increased with rising Shannon (*p* < 0.001) and Chao1 (*p* < 0.001) indices but declined beyond threshold values, yielding higher explanatory power (R^2^ = 0.23) than natural forests (R^2^ = 0.03). Conversely, natural forests showed a sustained positive correlation between NPP and Shannon diversity (*p* < 0.001) but no significant relationship with Chao1 (*p* = 0.55). These results highlight an asymmetric response of productivity to microbial diversity, with stronger mechanistic coupling in managed plantation forests.

### 2.3. Contrasting Microbial Community Structures and Functional Profiles

Natural forests showed significantly higher bacterial and fungal Shannon diversity than plantation forests (*p* < 0.01; Figure 3a). Bacterial Chao1 index was also greater in natural forests (*p* < 0.01), while fungal Chao1 peaked in plantation forests (*p* < 0.01; Figure 3b).

Distinct microbial community compositions emerged between forest types (Figure 3c). Bacterial assemblages in natural forests were dominated by nitrogen-cycling taxa (e.g., *Nitrobacter*, *Nitrospira*) and acid-tolerant genera (*Acidothermus*, *Candidatus Solibacter*), reflecting nitrogen transformation processes and acidic soils. Plantation forests, however, hosted bacteria linked to rhizosphere interactions (*Sphingomonas*, *Rhizomicrobium*) and pollutant degradation (*Arthrobacter*, *Flavobacterium*), alongside elevated plant pathogen loads (*Pseudomonas*, *Ralstonia*) (Figure 3c).

Beyond α-diversity, differences at the functional level were consistent with the observed productivity divergence (Figure 3d). Across the 237 observations, natural forests consistently exhibited higher productivity than plantation forests (Figure 3d). In natural forests, fungal communities included diverse functional guilds such as decomposers (*Inocybe*), mycorrhizal symbionts (*Cortinarius*), and nutrient-cycling taxa (*Archaeorhizomyces*). These guilds underpin nutrient acquisition and organic-matter turnover and tended to show positive associations with GPP and NPP. In addition, several bacterial taxa (e.g., *Sphingomonas*, *Flavobacterium*) also contribute to the decomposition of labile organic matter, further supporting nutrient cycling in natural forests. In contrast, plantation forests exhibited lower functional complexity and were enriched in ruderal/disturbance-tolerant and potentially pathogenic guilds, including taxa linked to pathogenicity and rhizosphere colonization, which displayed negative or more sensitivity-prone associations with productivity.

### 2.4. Drivers of Microbial Diversity: Climatic, Topographic, and Edaphic Controls

Figure 4a–p shows significant differences between natural and plantation forests in multiple climatic and edaphic factors (e.g., MAT, MAP, TN, pH; all *p* < 0.05), which likely drive the divergence of microbial communities, characterized by abundant genera, in both natural and plantation forest.

To elucidate the key environmental factors shaping microbial diversity in contrasting forest systems, we applied a Random Forest algorithm to identify the most influential predictors of bacterial Shannon and Chao1 indices in natural and plantation forest. In natural forests, the Shannon index was primarily influenced by MAP, PWQ, and POR (Figure 5a). For Chao1 index in natural forests, the model performance was poor, with only PS showing weak explanatory power (Figure 5b).

In contrast, plantation forest exhibited higher model performance for both diversity metrics. The Shannon index was best predicted by GRAV, longitude, and pH (Figure 5c). For the Chao1 index in plantation forest, CEC emerged as the dominant predictor, followed by TN and latitude (Figure 5d), underscoring the importance of nutrient buffering capacity and large-scale climatic proxies in shaping species richness under anthropogenic management regimes. These contrasting patterns highlight that microbial diversity in natural forests is more strongly regulated by climatic and biotic feedbacks, whereas plantation forests are primarily shaped by soil nutrient status and management-related edaphic factors. This divergence suggests that conservation strategies in natural forests should prioritize the protection of hydrological and plant-microbe interactions, while soil amelioration (e.g., enhancing CEC and nutrient balance) may be more effective for boosting microbial richness in plantation ecosystems.

### 2.5. Structural Equation Model Analysis

Structural equation modeling (SEM) revealed similar regulatory mechanisms governing primary productivity (GPP, NPP) in natural and plantation forests, driven by interactions among climatic factors, soil properties, and microbial diversity (Figure 6a,b). Generally, stronger causal relationships were observed in natural forests compared to plantation forests. Specifically, the major difference is that soil factors have a positive effect on microbial diversity (*SPC* = 0.298, *p* < 0.05), while negative values are found in plantation forests (*SPC* = −0.079, *p* < 0.05). In addition, microbial diversity shows a greater effect on GPP in natural forests (*SPC* = 0.214, *p* < 0.001) than in the plantation forests (*SPC* = 0.144, *p* < 0.05).

## 3. Discussion

### 3.1. Spatial Distribution, Climatic Context, and Diversity–Productivity Relationships

Natural and plantation forests in China exhibit clear spatial and climatic contrasts that underpin their differences in ecosystem functioning and productivity (Figure 1a,b). Natural forests are primarily distributed in mountainous and protected regions across northeast, southwest, and southern China, spanning a broad climatic gradient from cool–dry to warm–humid conditions. This broad distribution reflects long-term successional dynamics, structural complexity, and high ecological adaptability, enabling natural forests to maintain stable functions under diverse environments [15,16,17,18,19]. In contrast, plantation forests are concentrated in economically developed or ecologically restored regions of eastern and southern China, clustered within narrower climatic bands characterized by moderate temperature and precipitation. Their establishment through artificial planting and management—often emphasizing monoculture of locally dominant species—enables rapid canopy recovery but results in simplified community structures and reduced ecological stability [20,21]. This pattern highlights that natural forests sustain broader ecological resilience, while plantation forests remain more constrained by management priorities and climatic suitability.

Beyond spatial distribution, microbial diversity was strongly associated with forest productivity (Figure 2). In natural forests, NPP increased significantly with higher bacterial and fungal Shannon indices, reflecting the role of diverse microbial communities in supporting nutrient turnover, organic matter decomposition, and ecosystem stability [11]. In plantation forest, however, the relationships between NPP and microbial diversity indices (Shannon and Chao1) showed threshold-like responses: productivity initially increased with microbial diversity but declined at higher diversity levels. This pattern suggests greater sensitivity and instability under monoculture systems and anthropogenic disturbances. Reduced litter input and lower soil nitrogen availability under plantation forest further constrain microbial diversity and weaken its regulatory effect on productivity [20]. These contrasting patterns demonstrate that while climate establishes the baseline constraints on productivity, microbial diversity mediates resource-use efficiency and resilience. Natural forests, with higher microbial diversity and structural complexity, achieve more stable productivity across environmental gradients, whereas plantation forest remain functionally limited and more dependent on external management inputs [20,21,22].

### 3.2. A Comparative Study on the Differences in Microbial Community Diversity and Ecological Functions Between Plantation and Natural Forests

This synthesis reveals pronounced differences in microbial diversity and functional organization between natural and plantation forests across China. Natural forests exhibit higher microbial diversity and richness (Shannon and Chao1; Figure 3a,b), consistent with communities that are more dominated by resource-efficient, stress-buffered, and functionally persistent K-strategist life-history traits under relatively stable soil microenvironments [10]. In contrast, plantation forests show reduced diversity and a stronger imprint of monoculture management and anthropogenic disturbance, which tends to favor fast-growing and disturbance-adapted r-strategist taxa and can simplify belowground functional organization [23]. Under this framing, the key distinction is not only “how many taxa” occur, but also how community composition aligns with nutrient acquisition, organic matter turnover, and plant–microbe interactions that ultimately shape ecosystem functioning.

Beyond α-diversity, community composition indicates a shift in functional emphasis between forest types. In natural forests, the bacterial assemblage is enriched in taxa closely linked to nitrogen cycling and nutrient transformations, including *Bradyrhizobium* (often associated with N inputs) [24], nitrifiers such as *Nitrobacter* [25] and *Nitrospira* [26], and soil carbon–nutrient turnover–related groups such as *Candidatus Solibacter* [23]. These bacterial patterns align with a more internally regulated nutrient economy in natural forests. In parallel, fungal guild structure in natural forests suggests stronger representation of *K*-strategy functional groups that support sustained nutrient acquisition and organic matter turnover, including ectomycorrhizal fungi (ECM) [27], decomposers [28], and nutrient-cycling fungi such as *Archaeorhizomyces* [29]. Additional ECM- or nutrient-associated taxa, including *Dominikia* [30], *Amphinema* [31], and *Hygrophorus* [32], further support a functional profile consistent with more stable carbon and nitrogen cycling and tighter plant–soil feedbacks in natural systems.

By contrast, plantation forests are characterized by relatively lower functional complexity and a stronger signal of opportunistic or disturbance-tolerant functional strategies. Dominant bacterial genera include *Sphingomonas* [33] and *Rhizomicrobium* [34], which are frequently linked to rhizosphere interactions and stress tolerance, as well as taxa such as *Flavobacterium* [35] and *Gemmatimonas* [36] that participate in complex organic matter processing under fluctuating resource conditions. Plantation forests also show a broader representation of potential pathogen-associated or disturbance-responsive groups, including bacterial taxa such as *Pseudomonas* [37], *Bacillus* [38], *Rhodococcus* [39], *Ralstonia* [40], and *Streptomyces* [41], alongside fungal taxa such as *Alternaria* [27], *Curvularia* [42], *Trichoderma* [43], *Cladosporium* [44], *Aspergillus* [45], *Fusarium* [46], and *Penicillium* [47]. While some of these taxa may also include strains associated with pollutant degradation or plant growth promotion, their collective enrichment is consistent with a community more shaped by disturbance and simplified stand structure, potentially increasing biotic stress risks or weakening stable nutrient cycling relative to natural forests [48]. Importantly, these functional and life-history contrasts provide a mechanistic interpretation for the productivity divergence observed between forest types (Figure 3d): climate sets the baseline constraints, but differences in microbial functional organization and strategy composition can modulate how efficiently nutrients are acquired, recycled, and translated into productivity, thereby contributing to distinct productivity outcomes in natural versus plantation forests.

### 3.3. The Comprehensive Effects of Climate, Soil Factors, and Microbial Diversity on Forest Productivity

This study systematically disentangles the combined effects of climate, soil factors, and microbial diversity on forest productivity in China using multivariate correlation analysis and variance decomposition (Appendix A). Climate emerged as the dominant driver of both GPP and NPP (Appendix A), consistent with its well-established role in regulating global ecosystem functioning [49]. In both natural and plantation forests, temperature and precipitation directly shaped primary productivity, setting the fundamental baseline for ecosystem processes. However, the magnitude and pathways of soil and microbial contributions differed substantially between forest types. In natural forests, the availability of soil nutrients—particularly nitrogen—was strongly associated with stable productivity, indicating that nutrient supply is essential for sustaining growth [50]. By contrast, in plantation forests, the role of microbial diversity was weaker, likely due to simplified community structures and frequent human interventions such as fertilization and irrigation, which reduce the ecological functions of soil microbes [51]. Furthermore, the reduced diversity and weaker diversity–productivity relationships observed in plantation forest (Figure 2) indicate a trend toward microbial community homogenization, which diminishes their adaptive capacity and weakens their regulatory influence on productivity [52].

Mechanistically, our analyses highlight the complex interactions among climate, soil, and microbial communities. Soil fertility indicators, such as CEC and total nitrogen, played a pivotal role in buffering nutrient losses and enhancing plant nutrient uptake, with direct implications for productivity [12]. Microbial communities further modulated GPP by promoting nutrient mineralization, nitrogen fixation, and organic matter decomposition [53]. While their direct influence on NPP was weaker, microbes indirectly enhanced carbon allocation efficiency by sustaining soil nutrient cycling. The structural equation model revealed that climate not only exerts direct control on productivity but also indirectly shapes microbial activity via soil microenvironmental changes [54]. For instance, warming may accelerate decomposition of soil organic matter, temporarily increasing nutrient supply, but long-term depletion of soil carbon pools could lead to negative feedback on productivity [14]. Together, these findings underscore a hierarchical regulatory framework: climate sets the baseline, soil factors provide the nutrient context, and microbial diversity modulates the efficiency of resource acquisition and allocation, jointly determining forest productivity across natural and plantation systems. These findings also imply that practical steps can be taken to restore microbial diversity in plantation forest. Increasing tree species diversity, reducing intensive management practices, and applying organic amendments or microbial inocula may enhance microbial functional capacity and help stabilize productivity in plantation ecosystems.

### 3.4. Limitations and Future Directions

A key limitation is that natural forests and plantations in the compiled dataset may occupy partially different climate spaces (e.g., MAT/MAP ranges), which can confound synthesis-level cross-type comparisons because climate strongly constrains productivity and also shapes soil properties and microbial communities. Because the available literature does not provide sufficient climate-comparable natural–plantation sampling coverage across regions to support strict climate matching without substantially reducing representativeness, our findings should be interpreted as association patterns within the current data coverage rather than fully climate-controlled causal contrasts. Future work should prioritize climate-comparable paired designs and incorporate explicit management intensity variables to more robustly isolate microbial contributions. Overall, our study provides a unified climate–soil–microbe framework to interpret productivity divergence between natural and plantation forests under the current data coverage.

## 4. Method

### 4.1. Microbial Data Collection

To investigate how climate and soil properties influence microbial diversity and abundance in natural and plantation forests in China, a comprehensive database was developed. Relevant peer-reviewed articles published up to July 2024 were retrieved from Web of Science, Google Scholar, and the China National Knowledge Infrastructure (CNKI). The search strategy focused on peer-reviewed studies with titles, abstracts, or keywords containing terms such as (“forest” OR “soil microbial diversity” OR “bacterial diversity” OR “fungal diversity” AND “soil properties”), and restricted to those conducted in China. Because stand origin (natural vs. plantation forest) was rarely included in titles or keywords, we determined forest type by carefully examining the content of each study, including reported site history, stand management practices, and tree species composition. Notably, we did not use “planted forest” or “forest plantation” as search filters. Candidate studies were first retrieved using broad ecological terms, then stand origin was adjudicated during full-text screening based on site history, management records, and species composition reported by the authors. The retrieved literature was further screened according to the following criteria: (a) The study subjects included either natural or plantation forests. Natural forests refer to primary or secondary forests with minimal human disturbance, whereas plantation forests are human-managed and typically established for timber production or ecological restoration; (b) The study examined soil microbial diversity (bacterial or fungal) or its relationship with soil properties; (c) Soil properties (e.g., pH, organic matter content, fertility) and related variables were reported; (d) Data were complete and suitable for extraction and quantitative analysis.

Based on these criteria, studies that did not meet the inclusion requirements or lacked sufficient data were excluded, resulting in a final selection of 39 studies. The dataset comprised 237 observations, each representing a comparison between natural and plantation forests within the same geographic region or climatic zone to minimize large-scale confounding effects. The studies spanned a wide spatial distribution across China, including northeast, southwest, and southern regions, and covered diverse climatic gradients (MAT: –5 to 25 °C; MAP: 200–2000 mm) (Figure 1). The majority of studies provided single-year measurements, though the overall dataset reflects research conducted between 2013 and 2024. The database also contains detailed microbial diversity indices (e.g., bacterial and fungal Shannon and Chao1 indices), dominant genera, and their functional guilds. To evaluate microbial functional contributions, we assigned bacterial and fungal genera to functional guilds using the FUNGuild database and supporting literature [15]. Functional categories included decomposition, nitrification, nitrogen fixation, mycorrhizal symbiosis, pathogenicity, plant growth promotion, pollutant degradation, rhizosphere interaction, and entomopathogenicity. When relative abundance data were available, these values were used to quantify the contribution of each functional guild; when only qualitative records were reported, presence/absence information was considered. All microbial raw data were extracted from texts, tables, figures, and Appendix A of the included studies. Our dataset is a synthesis of site-/stand-level observations compiled from multiple studies rather than one-to-one paired natural–plantation comparisons. Therefore, cross-type contrasts are conducted at the synthesis level. To mitigate confounding, climate and soil covariates are explicitly incorporated in the SEM and machine-learning analyses; nevertheless, differences in climate-space coverage between forest types may remain and should be considered when interpreting results.

### 4.2. Environmental Data

Climatic variables (e.g., mean annual temperature [MAT], temperature seasonality [TS], maximum temperature of the warmest month [MTWM], annual precipitation [MAP], precipitation seasonality [PS], and wet-season precipitation [PWQ]) used in this study were obtained from the WorldClim global climate database (https://www.worldclim.org/, accessed on 1 July 2025) at a spatial resolution of 1 km. Soil properties (e.g., exchangeable potassium [AK], aluminum [AL], phosphorus [AP], cation exchange capacity [CEC], organic matter content [DC], soil moisture [DH], particle content [GRAV], pH, porosity [POR], and total nitrogen [TN]) for the 0–20 cm layer were sourced from the National Tibetan Plateau Data Center (https://cstr.cn/18406.11.Soil.tpdc.270281, accessed on 1 July 2025), with a spatial resolution of 1 km. This 0–20 cm depth corresponds to the topsoil layer most commonly reported in the included studies, and when studies used slightly different depths (e.g., 0–15 cm or 0–30 cm), we normalized them to the 0–20 cm equivalent to ensure comparability.

GPP and NPP data were extracted from the National Aeronautics and Space Administration (NASA; https://search.earthdata.nasa.gov/search, accessed on 1 July 2025), covering the period 2013–2024 at a spatial resolution of 250 m × 250 m. These data were derived from the Moderate Resolution Imaging Spectroradiometer (MOD13Q1) product and used within a light-use efficiency framework to obtain both GPP and NPP. Specifically, GPP was obtained using the C5 MOD17 algorithm, which has been widely validated against FLUXNET tower observations [16]. NPP is estimated using the Carnegie-Ames-Stanford Approach (CASA) model as follows:NPP(x,t) = APAR(x,t) · ε(x,t)(1)(2)GPP(x,t)=APAR(x,t) · ε_max · f(T,x,t) · f(W,x,t)
where APAR(x,t) represents the Photosynthetically Active Radiation (PAR, in MJ/m^2^) absorbed in the t-th month at pixel x, and ε(x,t) represents the actual light use efficiency (in g·C/MJ) in the t-th month at pixel x. εmax is the maximum light use efficiency under ideal conditions; and f(T,x,t) and f(W,x,t) are temperature and water stress scalars (0–1), respectively.

### 4.3. Data Analysis

This study explored the relationship between ecosystem productivity indicators (GPP and NPP) and various factors, including climatic factors (e.g., MAT, TS, MTWM, MAP, PS, and PWQ), soil factors (e.g., AK, AL, AP, CEC, DC, soil DH, soil GRAV, soil pH, soil POR, and TN), and microbial diversity (e.g., the Shannon index and Chao1 index for bacteria and fungi). Matrix correlation and network visualization techniques were applied to analyze these relationships. Correlations were calculated using the vegan package in R (version 4.3.1), and the relationships between variables and GPP/NPP were visualized using heat maps [17]. To quantify the relative contributions of different factors to GPP and NPP, multiple linear regression models were constructed, incorporating the aforementioned climatic factors, soil factors, and microbial diversity, along with an assessment of relative importance. The relative contribution of each predictor variable to the response variable was computed using LMG (Lindeman, Merenda, and Gold) with the relaimpo package in R. This approach assessed the independent explanatory power of each factor by decomposing the R^2^ values of the regression model, controlling for the effects of other variables. Furthermore, to analyze both the direct and indirect effects of each factor on GPP and NPP, piecewise structural equation modeling (SEM) was employed. Correlations were conducted on the R platform using packages such as piecewiseSEM, nlme, and lme4. The model’s goodness of fit was evaluated using Fisher’s C-test. A stepwise strategy, involving the deletion or modification of paths, was employed to optimize the model structure for a more rational causal explanatory framework. The model fit was assessed at a significance level of *p* < 0.05, with Fisher’s C/df between 0 and 2, and *p* values between 0.05 and 1.00. In SEM, the standardized path coefficients (SPC) indicate the direct effect of a variable assumed to be a cause on another variable assumed to be an effect. A greater SPC (close to 1) indicates that this effect is relatively high, and the plus or minus sign represents the positive or negative relationship between the two variables.

## 5. Conclusions

This meta-analysis demonstrates that climate variables set the primary baseline for forest productivity, but soil properties and microbial communities jointly modulate its efficiency and stability. Natural forests, enriched with diverse *K*-strategist taxa, maintain higher and more stable GPP and NPP, while plantation forests dominated by *r*-strategist and disturbance-tolerant taxa exhibit reduced functional complexity and greater sensitivity to environmental change. These findings highlight a hierarchical regulatory framework in which climate dominates, soil provides the nutrient context, and microbial functional guilds mediate productivity divergence between natural and plantation forests, offering insights for ecosystem modeling and sustainable forest management.

## Figures and Tables

**Figure 1 plants-15-00098-f001:**
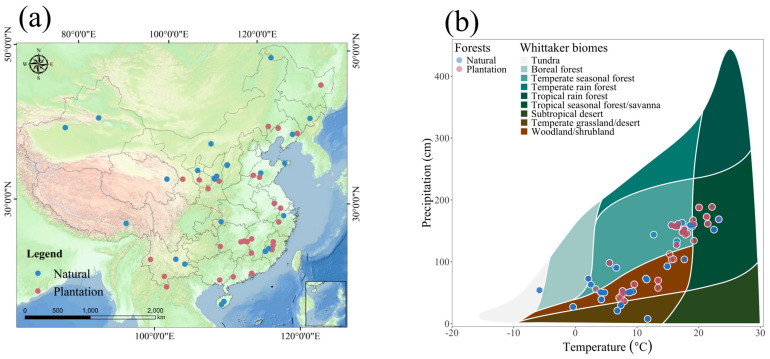
Spatial distribution of natural forests and plantation forests and their climatic conditions within Chinese regions: (**a**) Spatial distribution of NPP (g C/m^2^ year) in natural forests and plantation forests; (**b**) Distribution of Whittaker climate types for different forest types.

**Figure 2 plants-15-00098-f002:**
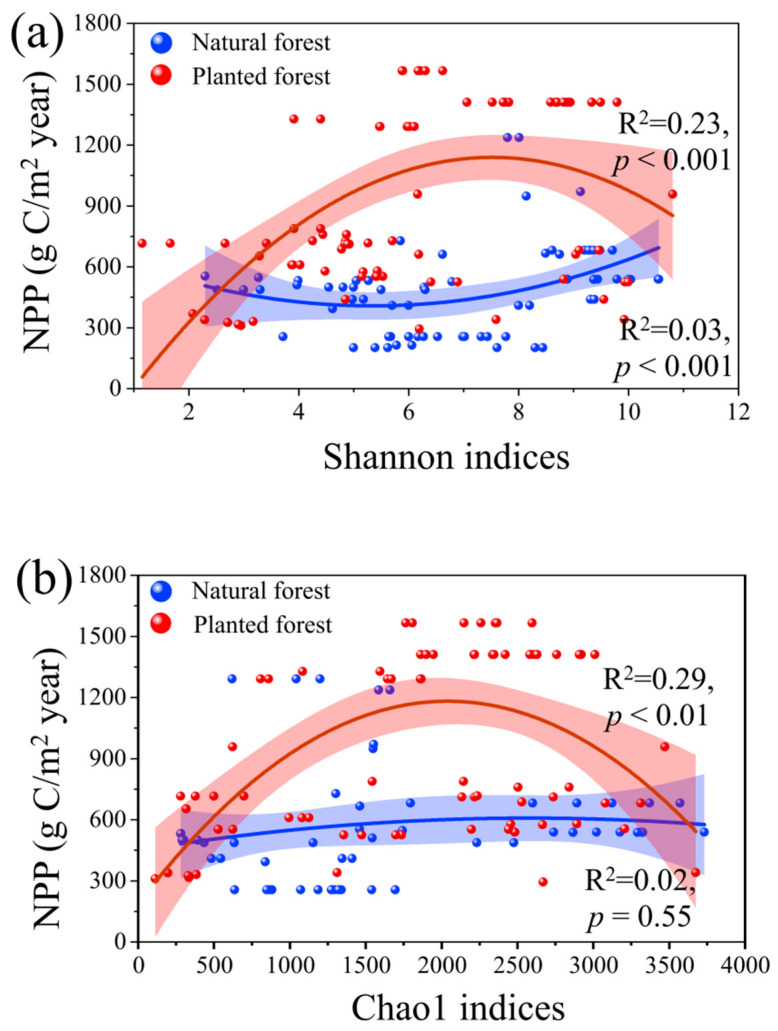
(**a**) Comparison of the relationship between Shannon indices with NPP (g C/m^2^ year) in natural and plantation forests; (**b**) Comparison of the relationship between Chao1 indices with NPP (g C/m^2^ year) in natural forests and plantation forests. *p* < 0.01 indicates significant differences.

**Figure 3 plants-15-00098-f003:**
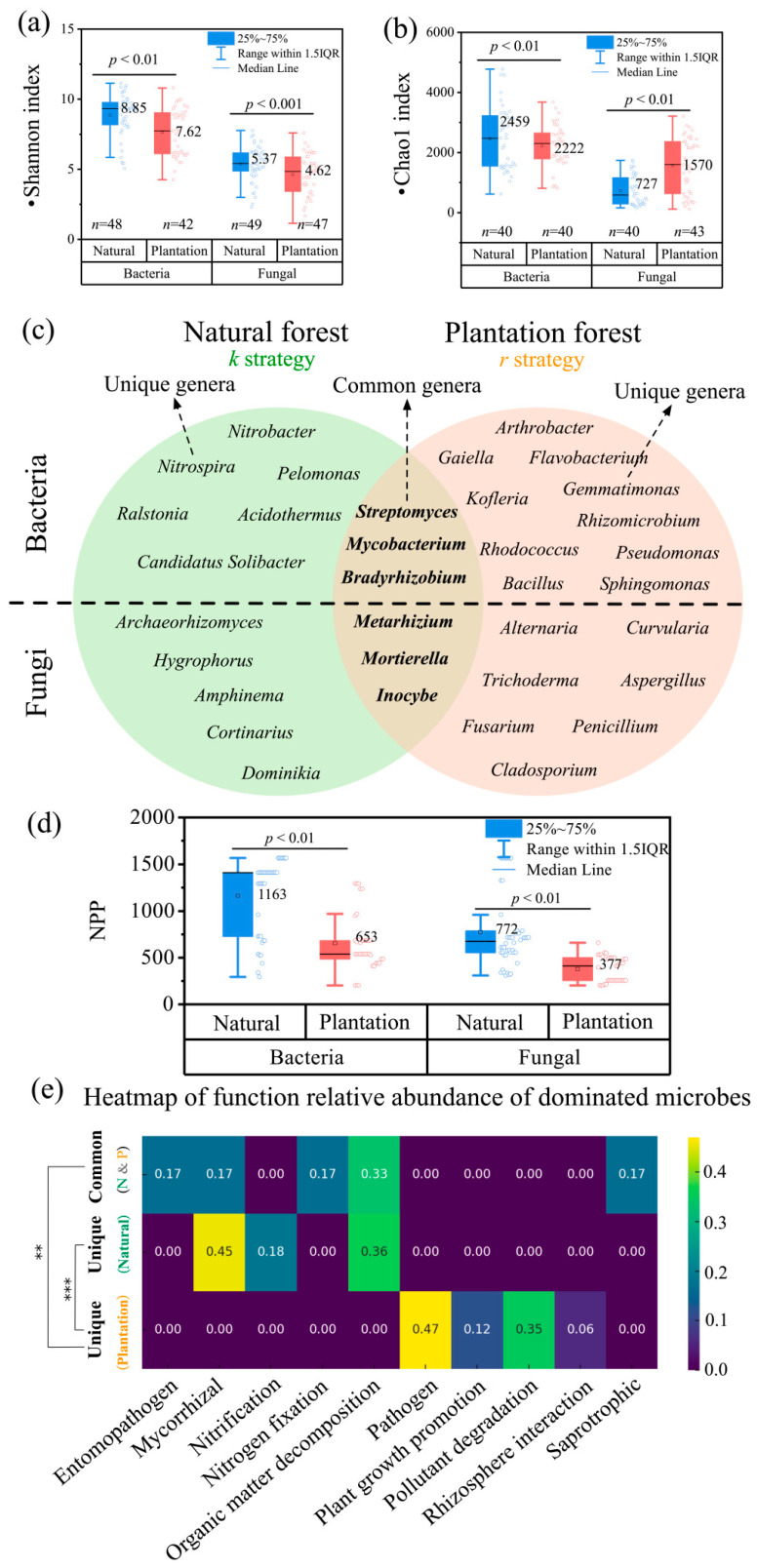
Comparison of (**a**) bacterial and fungal Shannon indices and (**b**) Chao1 indices between natural forests and plantation forests. n represents the sample size. (**c**) Venn diagram comparing dominant bacterial and fungal genera between natural forests (**left**) and plantation forests (**right**), with overlapping regions representing taxa common to both systems. Distinct microbial genera reflect ecosystem-specific functional traits linked to nutrient cycling and stress adaptation. (**d**) Comparison of NPP between natural forests and plantation forests in the bacterial dataset and the fungal dataset, respectively. (**e**) Heatmap of the relative functional abundance of dominant microbial communities. The first row illustrates the relative abundance of genera shared between both forest types, the second row highlights dominant genera exclusive to natural forests, and the third row features dominant genera unique to plantation forests.

**Figure 4 plants-15-00098-f004:**
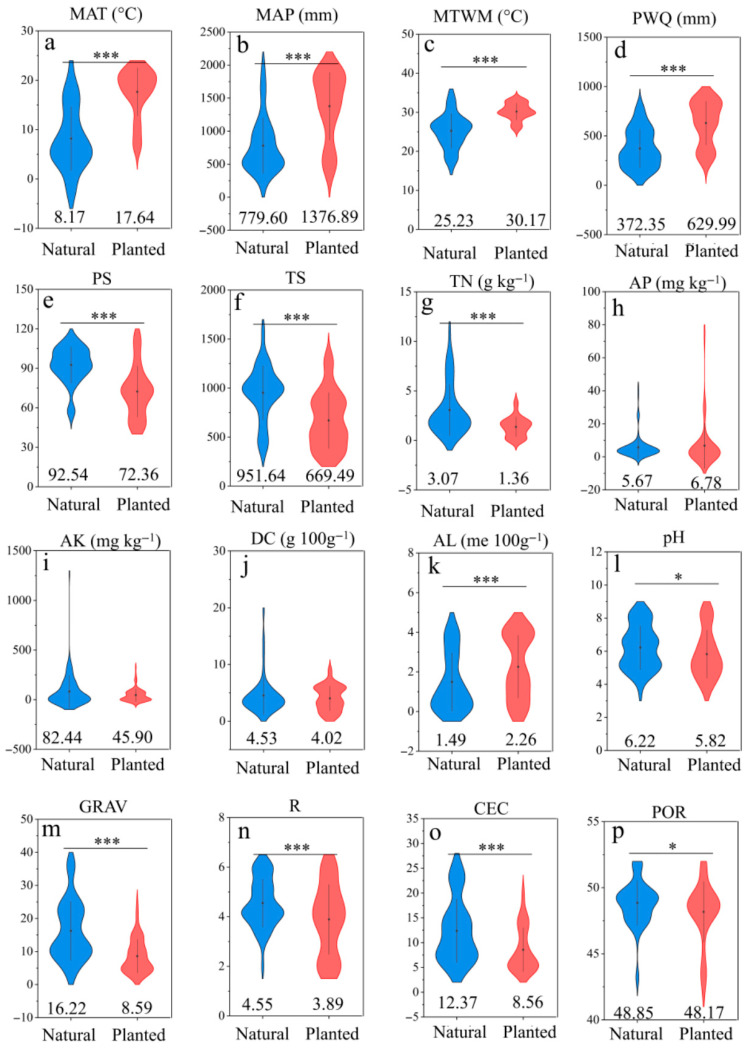
Comparison of the climatic and soil factors between natural forests and planted forests. Climatic factors include: (**a**) Mean Annual Temperature (MAT), (**b**) Annual Precipitation (MAP), (**c**) Max Temperature of Warmest Month (MTWM), (**d**) Precipitation of Wettest Quarter (PWQ), (**e**) Precipitation Seasonality (PS), (**f**) Temperature Seasonality (TS), Soil nutrient factors include: (**g**) Total Nitrogen (TN), (**h**) Available P (AP), (**i**) Available K (AK), (**j**) Dry Color (DC), (**k**) Exchangeable Al (AL), (**l**) Soil pH (pH), (**m**) Soil Particles (GRAV), (**n**) Root Abundance (R), (**o**) Cation Exchange Capacity (CEC), (**p**) Porosity (POR). The importance of these variables was estimated by the percentage increase in the Mean Squared Error (MSE, %). * indicates the significance level (*** *p* < 0.001; * *p* < 0.05). The value at the bottom indicates the mean value.

**Figure 5 plants-15-00098-f005:**
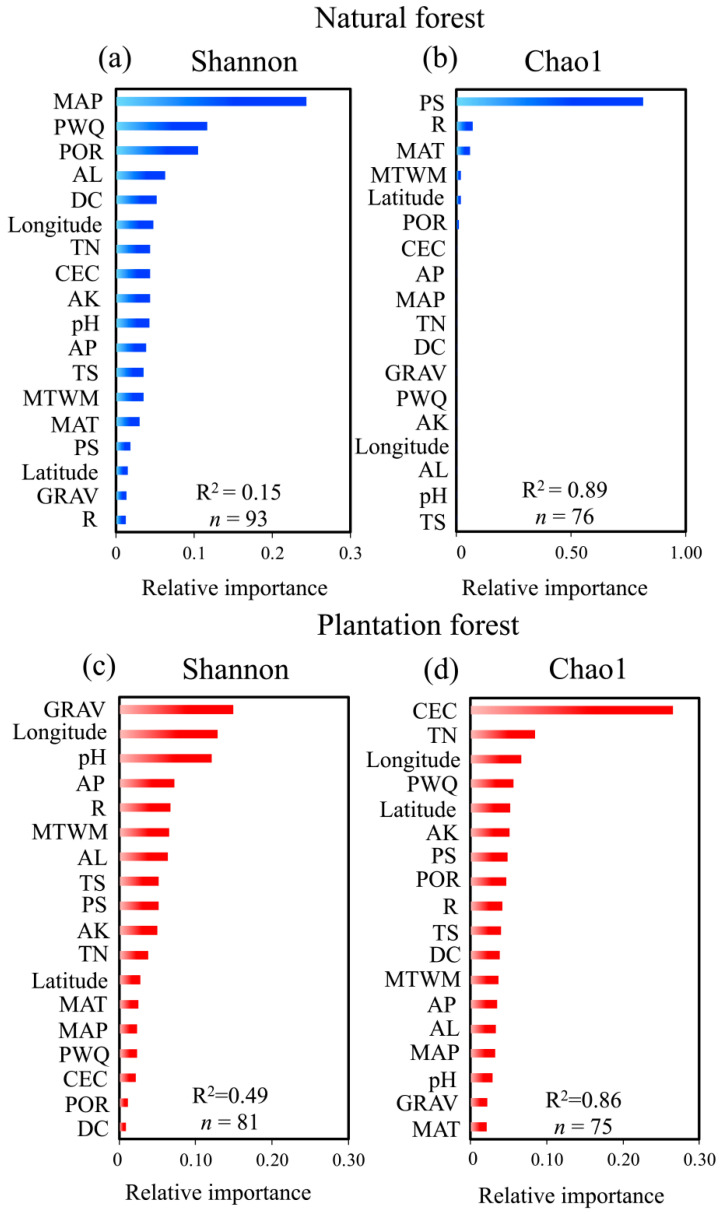
Relative importance of environmental predictors for microbial diversity in natural and plantation forests. (**a**,**b**) display results for the Shannon index and Chao1 index in natural forests, respectively. (**c**,**d**) display results for the Shannon index and Chao1 index in plantation forests, respectively. Predictors are sorted by descending mean importance within each panel. n represents the sample size.

**Figure 6 plants-15-00098-f006:**
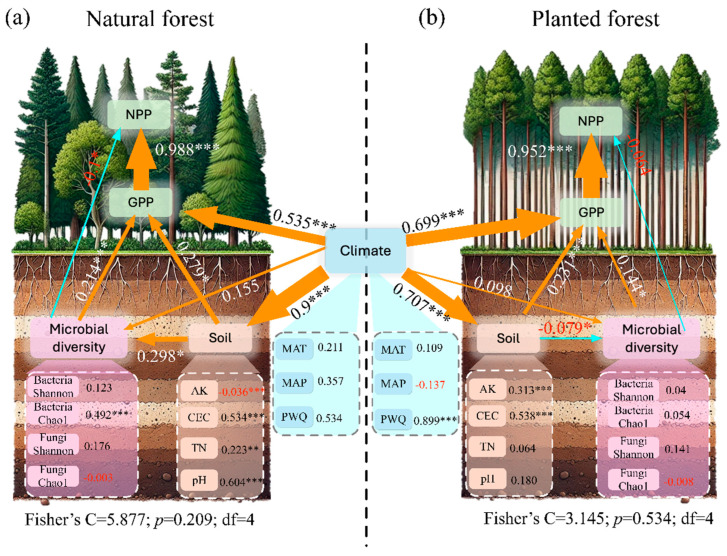
Relationships among climatic factors, soil properties, microbial diversity, and productivity in Chinese forest ecosystems. Structural equation models (SEM) evaluating direct and indirect drivers of gross primary productivity (GPP) and net primary productivity (NPP) in (**a**) natural forests and (**b**) plantation forests. Arrow direction indicates the direction of effect. Arrow width indicates effect size (*** *p* < 0.001; ** *p* < 0.01; * *p* < 0.05). The degree of freedom (df), Fisher’s C, and *p* value are shown in the lower middle.

## Data Availability

Data are available upon request.

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
