# Peer review of "Plants2026, 15(1), 98;https://doi.org/10.3390/plants15010098"

_plants, 2025, doi:10.3390/plants15010098_

Round 1

Reviewer 1 Report

Comments and Suggestions for Authors

A review on the manuscript titled “Soil Microbes Mediate Productivity Differences Between Natural and Plantation Forests”

The manuscript titled “Soil Microbes Mediate Productivity Differences Between Natural and Plantation Forests” is dedicated to the role of the soil microbial communities in natural and plantation forests. Using data analysis methods such as SEM and RFM, this study revealed specific differences between natural and plantation forests in the context of the soil microbiota influence on their productivity.

The intro of the study needs more work, and please provide the research hypothesis along.

I see some issues in the methodology (please see the specific comment at L221-222).

The text of the manuscript requires English proofreading to eliminate errors, misprints, etc.

The number of self-citations is quite high—6 out of 54.

The literature is up-to-date.

Specific comments:

L87-91. The authors cited the study with the same (237!) amount of observations as in the abstract and methods and postulated that “microbial diversity had a significant direct effect on gross primary productivity (GPP) in natural forests, but not in plantation forests”. Did they use the same dataset? Moreover, it sounds like everything is done with this research problem, taking into account the title of the manuscript. I suggest enhancing this paragraph in order to better explain the aim of their own study.

L217-222. Somewhere here should be a Fig. 2. Better do change the order of the figures.

L221-222. Maybe it is because the most amount of forest observations lie in the much less comfortable MAT and PRE (-5-+15°C, 10-100 mm) compared to the majority of the plantations (15-25°C, 50-200 mm) according to Figure 1. Data distributions do not coincide with the main climatic variables. You hardly can compare the microbial productivity of various forest types under such different climatic conditions; instead, it is more reasonable to put them under similar conditions. At Fig. 1, I can see a maximum of 10 forest-plantation pairs, which can be subjected to the comparison (25% of the involved studies). By this, you can exclude the most influential factor—climate—and get the more precise effect of microbes on the forest productivity.

L306. Consider increasing the resolution of the figure to enhance the visibility, especially of fonts.

L426-427. No need to introduce the CEC here again; it has been done before. Now use only abbreviations.

Author Response

Reviewer #1 (Remarks to the Author):

The intro of the study needs more work, and please provide the research hypothesis along.

Response: Thank you. We agree that the Introduction should be more focused and hypothesis-driven, ending with testable hypotheses that align with our questions and analytical framework.We add the  hypothesis according to your suggestion.Guided by these questions, we test the following hypotheses: H1, microbial diversity is more strongly and positively associated with productivity (GPP/NPP) in natural forests than in plantations within the current environmental coverage. H2, climate primarily sets the baseline productivity, while soil physicochemical and nutrient conditions shape microbial diversity and community structure; microbial diversity partially mediates soil–productivity linkages, and these mediation pathways differ between natural and plantation forests.

I see some issues in the methodology (please see the specific comment at L221–222).

Response: We appreciate the reviewer’s important methodological concern. We agree that our dataset represents a synthesis of non-paired observations across multiple studies and sites, and that natural forests and plantations may occupy partially different climate spaces (e.g., MAT/MAP ranges). This can confound cross-type comparisons and affect interpretation of microbial–productivity mechanisms. Given the compiled literature, we cannot implement strict climate matching without substantially reducing sample size and representativeness; therefore, we clarify in the revision that our analysis is a synthesis-level (non-paired) comparison and that inferences are primarily associational. We also explain in the Methods that we mitigate confounding by explicitly incorporating climate and soil covariates in the SEM/machine-learning framework, and we add a dedicated “Limitations and future directions” subsection stating that climate-space differences may bias estimated microbial effects and that causal interpretations should be cautious. Finally, we tone down causal/comparability wording in the Abstract (e.g., replacing “mediate/drive” with “are associated with/may contribute to” within the current data coverage) and highlight that future work should prioritize climate-comparable paired designs and/or incorporate management intensity information to more robustly isolate microbial contributions.

The text of the manuscript requires English proofreading to eliminate errors, misprints, etc

Response: We thank you for noting issues in English writing and presentation. We agree that the manuscript may contain grammatical errors, typos, and inconsistencies (e.g., articles, singular/plural forms, tense, terminology, abbreviations, variable names, and figure/table captions). In the revision, we will perform a systematic, full-text proofreading and consistency check, including standardizing abbreviation definitions at first mention, harmonizing terminology and variable naming across the main text and figures/tables, and ensuring correct formatting for taxonomic names (italics and capitalization). We will also revise key sentences/paragraphs to improve clarity and logical flow, followed by a final cross-check to ensure errors and misprints are thoroughly corrected.

The number of self-citations is quite high—6 out of 54

Response:Thank you for the careful check of our reference list. We re-checked all references and defined self-citations strictly as those that include any of the authors of this manuscript (Xing Zhang, Mengya Yang, Yangyang Liu, Jinkun Ye, Jiechen Tangyu, Jie Gao, Weiguo Liu, Yuchuan Fan). Our check indicates that only one reference overlaps with the author list, i.e., Ref. [16] (Zhang, X. … Gao, J., 2024, Forest Ecology and Management), whereas all other references do not include any of the manuscript authors. Therefore, we believe the statement “6 out of 54” may reflect a misunderstanding (e.g., confusion caused by common surnames).

L87-91. The authors cited the study with the same (237!) amount of observations as in the abstract and methods and postulated that “microbial diversity had a significant direct effect on gross primary productivity (GPP) in natural forests, but not in plantation forests”. Did they use the same dataset? Moreover, it sounds like everything is done with this research problem, taking into account the title of the manuscript. I suggest enhancing this paragraph in order to better explain the aim of their own study

Response: Thank you for pointing out that the paragraph in the Introduction (L87–91) could be misleading. We agree that mentioning the same sample size (237 observations) and stating the key SEM outcome (significant in natural forests but not in plantations) in the Introduction may confuse readers about whether an external study or the same dataset was used, and it may also give the impression that the core problem has already been solved in the Introduction. To avoid this confusion and to better clarify the aim of our study, we rewrote this paragraph. In the revised manuscript, the Introduction no longer reports any sample size or numerical/model-based results from our synthesis (e.g., “237 observations”, significance statements, or path coefficients). Instead, it now summarizes prior insights at a general level, highlights the remaining gap, and clearly states that our novelty is to integrate climate, edaphic conditions, and microbial attributes within a unified framework to compare mechanistic pathways underlying productivity differences between natural and plantation forests across China. All dataset details and effect sizes are confined to the Methods and Results sections.

L217–222. Somewhere here should be a Fig. 2. Better do change the order of the figures.

Response: Thank you for noting the mismatch between the in-text figure callouts and figure order/placement. We agree that the manuscript starts discussing content corresponding to Fig. 2 around L217–222, but the figure is not presented (or not numbered) in a way that aligns with its first mention, which disrupts readability. In the revised manuscript, we have harmonized the figure order and callouts by moving Fig. 2 to the location of its first citation (or renumbering figures accordingly), and we updated all in-text figure references and figure captions to ensure a consistent sequential order (Fig. 1 → Fig. 2 → Fig. 3 …).

L221–222. Maybe it is because the most amount of forest observations lie in the much less comfortable MAT and PRE (-5-+15°C, 10-100 mm) compared to the majority of the plantations (15-25°C, 50-200 mm) according to Figure 1. Data distributions do not coincide with the main climatic variables. You hardly can compare the microbial productivity of various forest types under such different climatic conditions; instead, it is more reasonable to put them under similar conditions. At Fig. 1, I can see a maximum of 10 forest-plantation pairs, which can be subjected to the comparison (25% of the involved studies). By this, you can exclude the most influential factor—climate—and get the more precise effect of microbes on the forest productivity.

Response: We appreciate the reviewer’s important methodological concern. We agree that our database is a synthesis of non-paired observations compiled across studies and sites, and that natural forests and plantations may occupy partially different climate spaces (e.g., MAT/MAP ranges), which can confound cross-type comparisons of microbe–productivity relationships. Because the underlying literature does not provide sufficient climate-comparable natural–plantation pairs, implementing strict climate matching at the revision stage would substantially reduce sample size and representativeness; therefore, we clarify that our analysis is a synthesis-level, non-paired comparison and that the resulting inferences are primarily associational within the current environmental coverage. In the revised Methods, we explicitly state this non-paired nature and describe how we mitigate confounding by incorporating key climate and soil variables as covariates/pathways in the SEM and as predictors in the machine-learning models. We also add a dedicated “Limitations and future directions” subsection noting that incomplete climate overlap may bias estimated microbial effects and that causal interpretations should be made cautiously. Finally, we have toned down causal/comparability wording in the Abstract and Discussion (e.g., replacing “mediate/drive” with “are associated with/may contribute to”), and we highlight that future work should prioritize climate-comparable matched/paired designs and, where possible, incorporate management intensity information to better isolate microbial contributions.

No need to introduce the CEC here again; it has been done before. Now use only abbreviations.

Response: Thank you for the comment; we removed the repeated definition of CEC and now use the abbreviation (CEC) consistently thereafter.

Reviewer 2 Report

Comments and Suggestions for Authors

This study aims to determine why natural and plantation forests in China differ in productivity by analyzing the interactions among climate, soil properties, and microbial diversity. By integrating data from 39 studies and applying advanced statistical methods, the research found that climate determines the primary level of productivity, soil provides the nutrient context, and microbial communities modulate resource use efficiency and ecosystem stability. The study reveals that natural forests have greater microbial diversity and functional structures, whereas plantation forests depend more on management and have functionally simplified microbial communities that limit their productivity.

The article is quite comprehensive and methodologically sound. A few comments:

The Methods section, according to the journal's requirements, should be the fourth section.

In the abstract: Please, emphasize the novelty of the research. In line 16, it is unclear what GPP and NPP specifically mean.

Introduction: The introduction is described in too much detail. Specific studies should not be mentioned, for example, in lines 47-50 and 57-63. It would be preferable to simply cite more literature sources related to the discussed issues. The paragraph in lines 66-72 stands out from the context. How is this related to the previous paragraph? There is no need to mention that you will use this in the study but justify how r/K strategies help to understand productivity differences. Please insert literature sources in the sentence ending at line 99.

Methods: How were your data from different sources was linked? Were the studies conducted under the same conditions? How were they standardized? Were the data transformed? Did you calculate NPP and GPP yourself (formulas 1 and 2), or were this data obtained from somewhere? Emphasize this.

The results are thorough and well presented.

Discussion: In lines 375-396, the names of bacteria are repeated that were discussed in the results (lines 244-257). In the discussion, it would be better not to repeat the same genera, but to group the bacteria according to their functions (e.g., nitrogen cycle bacteria, bacteria involved in mycorrhizae, pathogens, etc.). Also, include in the discussion what the limitations of the study were and what could be improved in the future, and emphasize the novelty of the study more.

Author Response

Reviewer #2 (Remarks to the Author):

The Methods section, according to the journal's requirements, should be the fourth section.

Response: Thank you for noting the journal’s formatting requirement. We have reorganized the manuscript so that the Methods section is now the fourth section.

In the abstract: Please, emphasize the novelty of the research. In line 16, it is unclear what GPP and NPP specifically mean.

Response: Thank you for the suggestion. We revised the Abstract to better highlight the novelty of this work by explicitly stating that our contribution is a synthesis-level, unified framework integrating climate, edaphic conditions, and soil microbial attributes to contrast mechanistic pathways underlying productivity divergence between natural and plantation forests. We also clarified the terminology by defining gross primary productivity (GPP) and net primary productivity (NPP) at their first mention in the Abstract and used these abbreviations consistently throughout the manuscript.

Introduction: The introduction is described in too much detail. Specific studies should not be mentioned, for example, in lines 47-50 and 57-63. It would be preferable to simply cite more literature sources related to the discussed issues. The paragraph in lines 66-72 stands out from the context. How is this related to the previous paragraph? There is no need to mention that you will use this in the study but justify how r/K strategies help to understand productivity differences. Please insert literature sources in the sentence ending at line 99.

Response: Thank you for the constructive suggestions regarding the Introduction. We agree that the previous version was overly detailed, that explicitly naming individual studies could dilute the main narrative, and that one paragraph was not well connected to the preceding text. Following your advice, we condensed and reorganized the Introduction by replacing case-by-case descriptions with synthesis-style statements supported by broader citations, and we rewrote/repositioned the previously disconnected paragraph to improve coherence along the climate–soil–microbe–productivity logic. We also integrated the r/K framework into the plantation-forest context and explicitly justified its relevance for interpreting microbial strategy shifts under disturbance/resource pulses and simplified belowground networks, thereby linking it more directly to productivity divergence.

Methods: How were your data from different sources was linked? Were the studies conducted under the same conditions? How were they standardized? Were the data transformed? Did you calculate NPP and GPP yourself (formulas 1 and 2), or were this data obtained from somewhere? Emphasize this.

Response: Thank you for these important methodological questions. We clarified in the revised Methods how observations from different sources were linked and standardized. Because the compiled studies were conducted under non-identical conditions, we first georeferenced each observation site using latitude/longitude and obtained productivity metrics from a unified source to ensure comparability. Specifically, GPP and NPP were obtained from NASA Earthdata (2013–2024; 250 m) by extracting gridded values at the coordinates of each observation site; GPP follows the C5 MOD17 light-use efficiency framework, and NPP is defined/estimated using the CASA formulation (Eqs. 1–2). We also added details on unit harmonization, temporal matching and quality control, and we mitigate potential confounding from between-study heterogeneity by explicitly incorporating key climate and soil covariates in the SEM and machine-learning analyses.

Discussion: In lines 375-396, the names of bacteria are repeated that were discussed in the results (lines 244-257). In the discussion, it would be better not to repeat the same genera, but to group the bacteria according to their functions (e.g., nitrogen cycle bacteria, bacteria involved in mycorrhizae, pathogens, etc.). Also, include in the discussion what the limitations of the study were and what could be improved in the future, and emphasize the novelty of the study more.

Response: Thank you for the suggestion on improving the Discussion. We agree that repeating the same genera listed in the Results reduces synthesis and mechanistic interpretation. In the revision, we reorganized the relevant section by grouping taxa by ecological functions (e.g., taxa linked to N cycling/nutrient transformations, taxa associated with mycorrhizal symbioses and nutrient acquisition, and potential pathogens or disturbance/opportunistic groups) and focusing on the underlying processes and implications for productivity divergence rather than re-listing genera. We also added/strengthened a dedicated “Limitations and future directions” subsection to explicitly acknowledge key constraints of this synthesis (e.g., cross-study heterogeneity, partial climate-space non-overlap, and limited management-intensity information) and to outline priorities for future work (e.g., climate-comparable matched/paired designs and incorporation of management metrics and refined spatiotemporal matching). Finally, we strengthened statements on the novelty of this study in the Discussion to clearly highlight our unified framework integrating climate–soil–microbe pathways to contrast mechanisms between natural and plantation forests.

L426-427. No need to introduce the CEC here again; it has been done before. Now use only abbreviations.

Response: Thank you for the suggestion. We agree that re-defining cation exchange capacity (CEC) in this part is redundant. In the revised manuscript, we removed the repeated definition at L426–427 and now use the abbreviation “CEC” consistently thereafter.

Round 2

Reviewer 1 Report

Comments and Suggestions for Authors

A review on the manuscript titled “Soil microbes mediate productivity differences between natural and plantation forests”

The manuscript "Soil microbes mediate productivity differences between natural and plantation forests" has undergone two rounds of revision. The authors have addressed the reviewer's comments and conducted a thorough proofread to correct grammatical errors, typos, and ensure overall consistency. These efforts have substantially enhanced the manuscript's clarity and logical flow, and I now recommend it for publication.

Author Response

Thanks.